# Advanced Characterization of Imiquimod-Induced Psoriasis-Like Mouse Model

**DOI:** 10.3390/pharmaceutics12090789

**Published:** 2020-08-20

**Authors:** Mehwish Jabeen, Anne-Sophie Boisgard, Alix Danoy, Naima El Kholti, Jean-Paul Salvi, Roselyne Boulieu, Bérengère Fromy, Bernard Verrier, Myriam Lamrayah

**Affiliations:** 1UMR 5305: Laboratoire de Biologie Tissulaire et d’Ingénierie Thérapeutique, CNRS/Université Claude Bernard Lyon 1, Institut de Biologie et Chimie des Protéines, 7 passage du Vercors, 69367 Lyon CEDEX 07, France; mehwish.jabeen@ibcp.fr (M.J.); boisgard.as@gmail.com (A.-S.B.); alix.danoy@ibcp.fr (A.D.); naima.elkholti@ibcp.fr (N.E.K.); Berengere.fromy@ibcp.fr (B.F.); Bernard.verrier@ibcp.fr (B.V.); 2UMR CNRS 5305, Pharmacie Clinique, Pharmacocinétique et Evaluation du Médicament, Université de Lyon, Université Lyon 1, 69373 Lyon CEDEX 08, France; Jean-paul.salvi@univ-lyon1.fr (J.-P.S.); roselyne.boulieu@univ-lyon1.fr (R.B.); 3Hospices Civils de Lyon, Groupement Hospitalier Edouard Herriot, Laboratoire de Biologie Médicale Multi Sites du CHU de Lyon, unité de Pharmacocinétique Clinique, 69002 Lyon, France

**Keywords:** psoriasis, imiquimod, mouse model, transepidermal water loss, angiogenesis, spleen, lymph nodes, imiquimod quantification in skin

## Abstract

Many autoimmune disorders such as psoriasis lead to the alteration of skin components which generally manifests as unwanted topical symptoms. One of the most widely approved psoriasis-like animal models is the imiquimod (IMQ)-induced mouse model. This representation mimics various aspects of the complex cutaneous pathology and could be appropriate for testing topical treatment options. We perform a thorough characterization of this model by assessing some parameters that are not fully described in the literature, namely a precise description of skin disruption. It was evaluated by transepidermal water loss measurements and analyses of epidermis swelling as a consequence of keratinocyte hyperproliferation. The extent of neo-angiogenesis and hypervascularity in dermis were highlighted by immunostaining. Moreover, we investigated systemic inflammation through cytokines levels, spleen swelling and germinal centers appearance in draining lymph nodes. The severity of all parameters was correlated to IMQ concentration in skin samples. This study outlines new parameters of interest useful to assess this model. We highlight the skin barrier disruption and report a systemic inflammatory reaction occurring at distance both in spleen and lymph nodes. These newly identified biological endpoints could be exploited to investigate the efficacy of therapeutic candidates for psoriasis and more extensively for several other skin inflammatory diseases.

## 1. Introduction

Nearly 3% of the world population suffers from inflammatory skin diseases such as atopic dermatitis and psoriasis, making them highly prevalent disorders known for their significant influence on the patient’s quality of life [1]. Psoriasis is a chronic topical disorder characterized by local inflammation, epidermal hyperplasia, leukocyte infiltration and increased vascularity in dermis [2]. This immune-mediated disease entails a dysregulated interplay between immune cells and keratinocytes. While the pathogenesis is not fully understood, intense research has been ongoing to explore the underlying mechanisms involved.

Several animal models mimicking human psoriasis have been developed and utilized over the years. In mice, the topical application of imiquimod (IMQ), a ligand of Toll-like receptors (TLR) 7 and 8, has been reported to induce psoriasis-like dermatitis [3]. It is a rapid and convenient model that allows the elucidation of underlying mechanisms and the evaluation of new therapies against psoriasis. It has been demonstrated that the IMQ-treated mouse model closely resembles human plaque-type psoriasis with respect to skin erythema, thickening, scaling, epidermal alterations (acanthosis, parakeratosis), and neo-angiogenesis, as well as to the inflammatory infiltrate consisting of T cells, neutrophils and dendritic cells [4]. The IMQ–treated mouse model is one of the most widely exploited models to study psoriasis [5]. In previously conducted studies, psoriasis area severity index (PASI) score, ear thickness measurements, histological staining, splenic involvement and the role of inflammatory cytokines have been very well established in this model [3,4,6]. This model has already been successfully exploited for ex vivo or in vivo screening of various conventional topical formulations [7] as well as novel drug nanovectors, such as poly(lactic acid) [8] or poly(lactic-co-glycolic acid) [9] nanoparticles, liposomes [10], lipid-carrier based gels [11] or β-cyclodextrin complexes [12]. However, there are certain parameters like skin barrier disruption, systemic inflammation, or the correlation between symptoms severity and skin concentration in IMQ that are not fully described for this model in the literature. The present study provides a deeper insight in the characterization of the IMQ-treated mouse model by exploring these parameters that have been well established in human psoriasis, unlike in mice, in addition to conventional ones.

It is well-known that the skin can be functionally and structurally altered by psoriasis. The impairment of skin barrier function is already described for psoriasis in mouse model [13]. However, there are not many studies that discuss the extent to which psoriasis actually modulates the skin barrier function [7]. There is a need to assess the changes in the cutaneous profiles of the intact and psoriatic skins. For this reason and in order to evaluate the disruption of skin barrier, the trans-epidermal water loss (TEWL) was measured in both phenotypes. TEWL is a non-invasive method that determines the amount of water lost (g/m^2^h) from the skin. In case of skin barrier disruption, the amount of water evaporating through the epidermis will increase, therefore increasing the TEWL values. Thus, this parameter can be used as a standard to measure the integrity of skin barrier [14,15].

Vascular endothelial cells are also known to be active participants in the psoriatic inflammatory process [16]. The platelet endothelial cell adhesion molecule PECAM/CD31 is highly expressed at the lateral junctions of endothelial cells in existing and newly formed blood and lymphatic vessels, making it a useful and commonly evaluated endothelial cell marker [17,18,19]. In addition, PECAM/CD31 plays a role in angiogenesis, platelet aggregation, homeostasis, and in the maintenance of vascular endothelial barrier function [19]. As a second marker, the presence of CD34 positive endothelial cells has been observed in psoriasis since this transmembrane protein is involved in the adhesion phenomenon of blood vessels [20,21,22,23]. Thus, we used both anti-CD31 and anti-CD34 staining to confirm the usefulness of these parameters in the IMQ mouse model for the study of angiogenesis and dermal vascularity changes.

Next, phosphorylated signal transducer and activator of transcription 3 (pSTAT3) is one of the key players in the pathogenesis of psoriasis [24,25,26]. It is considered to be an important target for the development of psoriasis therapies as it has been found to be active on psoriatic keratinocytes [27,28]. Miyoshi et al. reported that the inhibition of STAT3 is effective against psoriatic phenotype occurrence in transgenic mice with constitutively active STAT3 (K5.Stat3C) as well as in psoriatic patients [29]. Thus, the staining of this marker could be a useful indicator of the progression of psoriasis in the IMQ-induced mouse model and subsequently for a potential screening of therapeutic inhibitors.

Another well-documented fact regarding psoriasis is the systemic inflammatory process occurring in the lymph nodes and spleen. As a secondary lymphoid organ, the spleen is an important marker to study the systemic inflammatory process and immune response [30]. Similarly, the importance of lymph nodes in immune mediated diseases such as psoriasis cannot be ignored. To our knowledge, there has been several studies on the usage of these secondary lymphoid organs as inflammatory markers but only a few were able to correlate these changes with the severity of inflammation in psoriasis. Here, in order to determine the possibility of this correlation to be a marker of psoriasis severity, the changes in spleen and lymph nodes were reported along with the staining of germinal centers.

Above all, the IMQ quantification in the skin samples using high performance liquid chromatography (HPLC) technique adds to the originality of this research work. This aimed to relate the outbreak profile of the monitored parameters with the absorbed concentration of the drug in skin. Although the IMQ-induced mouse model is the most well-established tool to study psoriasis, this is the first study that correlates the IMQ concentration with the progression of psoriasis. This study shows the usefulness of the chosen parameters, which include: (i) the extent of skin barrier damage, (ii) the systemic inflammation, (iii) the expression of vascular endothelial markers and (iv) the inflammatory changes in lymph nodes related to quantification of IMQ in skin. Ultimately, this paper highlights the relevance of controlling these novel parameters when studying psoriasis pathogenesis and, consequently, when developing novel pharmaceutical therapeutics.

## 2. Materials and Methods

### 2.1. Animals

Twenty male BALB/cByJ mice between 7 to 8 weeks old were purchased from Charles River Laboratories (L’Arbresle, France) and housed at the Plateau de Biologie Expérimentale de la Souris (PBES, ENS Lyon, France) for seven days prior to experimentation. All animals were handled according to the institutional regulations and guidelines. The study and procedures were approved by the Ethical Committee of Rhône-Alpes for the Animal Experimentation (CECCAPP, Lyon, France) under the identification code ENS_2014_033, approved on 20 June 2014.

### 2.2. Imiquimod Induced Psoriasis

The same protocol as employed by van der Fits et al. [4] was followed for the induction of psoriasis-like lesions in mice. Briefly, the mice were shaved and depilated on their back using a Brav Mini Clipper razor (Wahl, Sterling, ILL, USA) and commercially available depilatory cream (Veet^®^, Reckitt Benckiser Group, Slough, England). The experimental group was treated daily with application of 62.5 mg of commercially available 5% IMQ cream (Aldara, MEDA Pharmaceuticals, Solna, Sweden) on the depilated back and right ear of each mouse. Thus, the total mass of IMQ distributed daily to each mouse is 3.125 mg. This dose has been empirically determined [4] and already demonstrated to cause reproducible and optimal skin inflammation in this model in various publications [7,27,31,32,33]. The control group was treated similarly with Lanette cream containing cetyl stearyl alcohol (BASF, Ludwigshafen, Germany). The animals were divided into five groups (*n* = 4), including three experimental mice and one control mouse, for each day of euthanasia (day 2, 3, 4, 6 and 8).

After sacrificing the mice, skin ear samples, spleen and both inguinal lymph nodes were collected. Skin and ear sampled were then fixed in 4% *v*/*v* paraformaldehyde diluted in phosphate buffer saline overnight, followed by paraffin embedding. Five μm sections were then cut using a microtome (Rotary Microtome Model 1212, Leica, Wetzlar, Germany). For each mouse, an 8 mm skin sample was also harvested using a sterile skin punch and immediately flash-frozen in liquid nitrogen and then stored at −80 °C.

### 2.3. Quantification of Imiquimod in Skin

IMQ was extracted from mouse skin using a protocol adapted from De Paula et al. [34]. Briefly, skin samples were shredded in an extraction medium composed of 70% methanol–30% ammonium acetate buffer (pH 4.5, 100 mM). Samples were then sonicated thrice for 30 s each using an ultrasound probe (FB15060, Fisher Scientific, Hampton, NH, USA). Finally, samples were filtered through a 0.40 µm pore-size membrane filter and stored at 4 °C until HPLC analysis.

The HPLC system consisted of a 2795 separation module (Waters, Agilent, Santa Clare, CA, USA) a Waters 2996 UV-DAD detector and the Waters Empower software system. Chromatographic separations were carried out on a Kinetex^®^ 5 µm C18 100Å (150 × 4.6 mm, Phenomenex, Torrance, CA, USA). Elution was performed using a gradient of two mobile phases. Phase A consisted of acetate ammonium buffer (5 mM, pH 3.85) and phase B was acetonitrile. Gradient started with 80% phase A during 6 min, then decreased from 80 to 65% for 1 min and remained at 65% from minute 7 to 16. After 16 min, the gradient increased back to 80% of phase A for 1 min and remained at 80% for 3 min. Flow rate was 0.7 mL/min, the volume of injection was 20 µL and the analyses were carried out at room temperature (RT). The analysis runtime was 20 min and the detection wavelength was set at 254 nm.

The described procedure was validated according to ICH guidelines (Topic Q2B, 1996) using spiked samples of different concentrations of IMQ in skin extracts. The calibration curve was linear in the concentration range from 0.5 to 100 µg/mL. The regression line equation calculated by the least-squares method was y = 164,548.7x + 7909.5 with a correlation coefficient (r) of 0.9999 and slope RSD (*n* = 3) of 3.23%.

### 2.4. Psoriasis Area Severity Index (PASI) Score Evaluation

The PASI clinical scoring system was used to assess the inflammatory status of the mice dorsal skin for all the 8 days. It included the visual examination of the following three parameters: erythema (redness), induration (thickness) and desquamation (scale) on the back skin of each mouse. Each parameter was given a score between 0 and 4 (0-none, 1-slight, 2-moderate, 3-marked, 4-very marked) leading to a cumulative score from 0 to 12. The evaluation was done independently by two researchers (*n* = 6 for each parameter) and the mean of values was then calculated.

### 2.5. Ear Thickness Measurements

For all the 8 consecutive days, the ear thickness measurements for both ears of each mouse were performed using a micrometer (Türlen Anytime Tools, Granada Hills, CA, USA) in µm and the mean of values was calculated (*n* = 15 for day 1 and decreased subsequently to *n* = 3 for day 8).

### 2.6. Histology

The dorsal skin and ear samples of each mouse were stained with Masson’s Trichrome staining to evaluate the detailed histological features of psoriasis. Briefly, the skin and ear samples were deparaffinized and the staining was performed using the hematoxylin solution, mixture of orange-G, ponceau acid fuchsin and light green coloring agent using the standard procedure. The samples were then observed under a light microscope (Eclipse Ti-E inverted microscope, Nikon, Minato-ku, Japan) and obtained images were analyzed using ImageJ software version 1.53c (NIH, Bethesda, MD, USA).

For each section, a representative area was selected in order to assess inflammatory aspects of the skin based on both epidermal thickness and inflammatory cells infiltration. The epidermal thickness values for all the three mice of each day was measured using ImageJ software through area and length measurements. The same procedure was repeated for the control group. The measurements were done independently by two researchers (*n* = 6) and statistically analyzed by one-way ANOVA followed by Tukey multiple comparison post hoc analysis. One randomly selected picture per day has been presented in this paper.

### 2.7. Phosphorylated Signal Transducer and Activator of Transcription 3 (pSTAT3) Staining

The overexpression of pSTAT3 in the psoriatic skin was determined by immunofluorescence. The skin samples were then stained with primary antibody anti-pSTAT3 (Tyr705) (1:100, Cell Signaling Technology, Danvers, MA, USA) following the standard procedures and localized by a secondary anti-rabbit antibody Alexa-Fluor 647 (1:500, Life Technologies, Carlsbad, CA, USA). The sections were also stained with DAPI (1:100, Vector Laboratories, Burlingame, CA, USA) before mounting. The stained slides were observed under a fluorescence microscope (Eclipse Ti-E inverted microscope, Nikon, Minato-ku, Japan). One representative image per day has been shown in this paper.

### 2.8. Trans-Epidermal Water Loss (TEWL)

The skin barrier function of each mouse was assessed before euthanasia. Following mouse anesthesia using isoflurane, the measurements of TEWL were recorded in g/m^2^h using the Aquaflux device (Biox Systems Ltd., London, UK). The areas considered for the measurements included back, both flanks and both ears. The evaluation was done independently by two researchers (*n* = 10) and the mean of values was then calculated. The values for the affected and control areas were compared and statistically analyzed using a one-way ANOVA coupled with Dunnett multiple comparison post hoc analysis.

### 2.9. Vascular Endothelial Markers Staining

The expression of the vascular markers CD31 and endothelial cells CD34 was assessed by immunofluorescence staining using the rabbit anti-CD31 (1:50, ab28364, Abcam, Cambridge, UK) and the rat anti-CD34 (1:50, ab8158, Abcam) primary antibodies. The formalin preserved skin samples of the psoriatic and naive mice were stained. Sections of 5 µm thickness were cut, deparaffinized in methylcyclohexane, and rehydrated in solutions of graded ethanol. Sections were then immersed in citrate buffer 10 mM citric acid (pH 6) at 95 °C for 30 min to unmask the antigenic sites and followed by the immersion in blockaid (ThermoFisher Scientific, Waltham, MA, USA) at RT to block the non-specific binding sites. Sections were then incubated with primary antibodies at 4 °C overnight and subsequently treated using the secondary antibody mix including anti-rabbit Alexa-Fluor 647 (1:500, Life Technologies, Carlsbad, CA, USA) and anti-rat FITC (Jackson Immunoresearch, Cambridge, England). The sections were also stained with DAPI (1:100, Vector Laboratories) before mounting. Negative controls consisted of adjacent sections from each sample that were processed without the primary antibodies. The analysis of stained sections was performed using a fluorescence microscope (Eclipse Ti-E inverted microscope, Nikon). The images were then treated using ImageJ and one random image per day was selected.

### 2.10. Quantification of Inflammatory Cytokines in Serum and Skin

Blood was collected using the retro-orbital route from the anesthetized mice (using isoflurane) before euthanasia. Samples were then centrifuged twice for 10 min at 16,000× *g* and supernatants were preserved after each centrifugation. The final supernatant obtained was stored at −80 °C until further analysis.

Protein extraction was performed on frozen skin biopsies following the protocol described by Amsen et al. [35]. Briefly, skin samples were crushed in the lysis buffer using the Eppendorf fitting pestle (Fisher Scientific, Hampton, NH, USA),incubated at 4 °C and then sonicated for 10 s thrice. The supernatant was extracted after the centrifugation at 19,000× *g*, for 15 min at 4 °C and then stored at −20 °C before further analyses.

Quantification assay for the following inflammatory cytokines interleukin IL-6, IL-1β, IL-17A and TNF-α was performed on both skin and serum samples using the ELISA kits (ELISA Ready-SET-GO, eBioscience, San Diego, CA, USA) according to the manufacturer’s instructions.

The statistical difference between each day was determined using one-way ANOVA followed by Tukey multiple comparison post hoc analysis where, *n* = 3 for experimental group of each day and *n* = 4 for control group.

### 2.11. Spleen Length and Mass

Following the euthanasia of mice, the spleens were carefully removed. Apart from visual examination, the length of each spleen was determined using a scale in cm and the spleen weight was recorded in g using a digital balance (AE163, Mettler, Columbus, OH, USA).

The measurements are expressed as mean ± SD (*n* = 3 for experimental group and *n* = 4 for control). The values for the affected and control mice were then statistically analyzed using one-way ANOVA test followed by Dunnett multiple comparison post hoc analysis.

### 2.12. Lymph Nodes Staining

The two inguinal lymph nodes from each mouse were harvested, placed in Optimum cutting temperature medium (OCT) and immediately flash frozen in liquid nitrogen for germinal center staining. The OCT frozen lymph nodes were cut in 6 μm sections using a cryostat (CM 3050S, Leica, Wetzlar, Germany). After mounting the sections on slides, immunofluorescence staining was performed following the protocol given by Mastelic Gavillet et al. [36] and Lofano et al. [37]. Briefly, the fixation of slides using acetone was followed by the blocking of non-specific binding sites using blockaid (ThermoFisher Scientific, Waltham, MA, USA) at room temperature (RT) for 30 min. The slides were then incubated for 30 min using the primary antibody mix, consisting of purified anti-mouse IgD (1:200, Biolegend, San Diego, CA, USA) and PNA biot (1:100, Biotinylated Peanut Agglutinin). IgG goat anti-rat AF488 and Cy3-Streptavidine (1:100, Life Technologies) were then used as secondary antibodies. To confirm the staining of germinal centers, negative control from each sample was processed without the primary antibody mix. Finally, the slides were mounted with the Vectashield mounting medium without DAPI (Vector Laboratories) and were observed under fluorescence microscope (Eclipse Ti-E inverted microscope, Nikon).

The images of all stained lymph nodes were then analyzed using ImageJ software. The measurements of the area, length and width of each lymph node was made and compared *n* = 3 for experimental group of each day and *n* = 4 for control group and statistically analyzed by one-way ANOVA followed by Dunnett multiple comparison post hoc analysis. The number of germinal centers for each lymph node was also noted. One representative image was selected for each day.

### 2.13. Statistical Analysis

Results are expressed as the mean ± SD whereas, for each day *n* = 3 for the experimental group and *n* = 1 for the control group for each day of euthanasia. Data was statistically analyzed using GraphPad Prism Version 7.0 software (San Diego, CA, USA). The statistical difference between groups was analyzed using one-way ANOVA tests followed by specific multiple comparison post-hoc tests as mentioned with each observed parameter individually.

## 3. Results

### 3.1. Relation of Imiquimod Concentration to the Severity of Symptoms

The results obtained from HPLC showed that the IMQ is quantified in skin from day 2 around 100 µg/g. An increase is observed at day 6 to 8 where the concentration reached 600 µg/g (Figure 1).

### 3.2. Severity of Skin Inflammation

The signs of erythema, thickness and scales on the back skin were seen two or three days after the application of IMQ. All the three parameters along with the cumulative score showed the gradual increase in the observed physical factors with the maximum score at day 8 as compared to the control group (Figure 2, up and down). One mouse died during the adaptation week so there was no control for day 2.

### 3.3. Ear Thickness Measurements

The application of IMQ on the right ear for 8 consecutive days caused a continuous increase in thickness starting from day 5 until the end of the experiment due to the inflammation process (red squares, Figure 3). Contrarily, on the left ear, control ear without treatment, no change in ear thickness was observed for 8 days (red rounds). For the control group, both right (blue squares) and left (blue rounds) ear measurements were done, and no change was observed for 8 days.

### 3.4. Increased Epidermal Thickness and Infiltration of Keratinocytes

The histological observation of the Masson’s trichrome stained sections of ear and dorsal skin confirmed the results observed with the PASI score and ear thickness measurements. At the microscopic level of IMQ-treated skin and ears, there was an increase in the epidermal thickness and infiltration of keratinocytes. However, the control group showed an intact skin with no signs of thickness and infiltration (Figure 4).

### 3.5. Expression of pSTAT3 Proteins

The psoriatic skin slides stained with anti-pSTAT3 clearly reveals the signal for pSTAT3 proteins found in the basal layer of epidermis on days 6 and 8. On the contrary, they are not expressed in the control skin samples (Figure 5).

### 3.6. Skin Barrier Disruption

The experimental results for TEWL measurements of the back showed identical results from three different sites including left flank, right flank and upper back. A significant increase of TEWL values from 20 g/m^2^/h to approximately 80 g/m^2^/h was observed starting from day 4 of IMQ application. This result indicates the disruption of the epidermal barrier caused by psoriasis overtime. Likewise, the TEWL measurements for the right ear (treated with IMQ) showed a significant increase, whereas, the TEWL values for the left control ear (without IMQ) did not increase statistically during all the experimental days (Figure 6).

### 3.7. Dermal Hypervascularity and Angiogenesis

The immunofluorescence staining of vascular and endothelial markers started appearing in the dermal layer from day 4 and was observed in the days 6 and 8 as well (Figure 7). The expression of their signal at these days can be very well related to the literature which reports increased dermal vascularity and angiogenesis in the pathogenesis of this disease.

### 3.8. Inflammatory Cytokines

Cytokines produced by T cells are predominant mediators of skin pathology in psoriasis. Increased cytokines concentration was observed in case of IL-6, IL-1β and Il-17A in skin samples. In blood samples, increased concentration was observed only with IL-17A and TNF-α. However, only the increase of TNF-α in blood at day 2 was significant (Figure 8).

### 3.9. Systemic Inflammatory Response

The observation of the secondary lymphoid organ, spleen showed an increase in the mass and length with the progression of psoriasis. The length of the spleen showed a significant increase at day 2 and the difference with the control length became very significant (*p* < 0.0001) with the progression of days. In case of splenic mass, the data showed a significant change starting from day 3 and a total of four-fold increase at day 8 (Figure 9B). Furthermore, a similar trend was seen in case of the area of lymph nodes. The statistical data showed a three-fold increase in the total area (Figure 10). The data is presented as mean values for the respective sampling days. More importantly, the appearance of germinal centers in all the treated mice at day 8 signifies the fact that this immune mediated response of lymph nodes can be related to the severity of psoriasis and is in compliance with other results (Figure 11).

## 4. Discussion

Despite the development of numerous preclinical models to assess the different aspects of topical and transdermal delivery, the complexity of skin barrier in terms of structure and composition remains a limiting factor for drug penetration, even in diseased skin [38,39]. The IMQ-induced psoriasis mouse model recapitulates several features of skin inflammatory diseases similar to human psoriatic inflammation. These characterized features include increased thickening of the epidermal layer, elongation of rete-like ridges and hyperkeratosis after IMQ treatment. Our study showed all these parameters which are in agreement with previous studies [4,6,37].

For the first time, this study focuses on monitoring the quantity of IMQ found in the skin following its topical application. The molecule is detected in skin after two days of daily application and its concentration increased three folds on the day 6 of experimentation. This kinetic of penetration and retention can be explained by the physical and chemical properties of IMQ. The low molecular weight (240.3 g/mol) allows an easy skin penetration but due to its lipophilic properties (logP = 2.6) the molecule is mainly retained in the stratum corneum. The increase in skin drug concentration is related with skin changes that appeared from day 2 to 3. The total PASI score highlights the psoriasis severity notably through the observation of desquamation representing the scale formation. Layers of stratum corneum are detached and thus, promote better IMQ skin penetration.

In parallel, a significant epidermis thickening is observed both on the back and on the treated ear starting from day 4 and increasing throughout the experiment. This epidermal hyperplasia is correlated with the appearance of pSTAT3 that is a key transcription activator involved in the development of psoriatic lesions [27]. The histological analysis of pSTAT3 on the psoriatic skin slides is clearly consistent with the results shown in the previous studies that elucidate the expression of this protein in the keratinocytes of psoriatic skin [34,36]. Furthermore, these findings are in agreement with Sun et al. [27] where they proved berberine to be an effective therapeutic option for psoriasis as it targets pSTAT3. Altogether, these insights show that pSTAT3 staining is a relevant marker to successfully evaluate future anti-psoriatic drugs using IMQ-induced mouse model.

The skin is a semi-permeable barrier which not only regulates the penetration of exogenous compounds from the outside, but also prevents the loss of intrinsic water in the environment. This phenomenon is possible due to the keratinocytes differentiation that generates epidermal barrier that prevents water loss [40]. These findings support our result where increased trans-epidermal water loss in psoriatic skin is seen as compared to the control skin. In accordance with the evolution of the previous presented parameters, TEWL measurements on the back and treated ear increased from day 3 to 4 to reach a maximum at day 6 to 8 where PASI score and epidermal hyperplasia are maximal. The results in our study are also analogous to the studies shown by Lin et al. [7] and Shah et al. [33]. Furthermore, this parameter describing the severity of damage to the skin barrier [10] reached a significant high value after 4 days of IMQ application like it was described by Takahashi et al. [41]. Moreover, this can also be supported by circumstantial evidence that states that psoriatic inflammation is characterized by acanthosis, hyperkeratosis and infiltration of inflammatory cells into epidermis, ultimately leading to the formation of erythematous plaque, and resulting in the loss of the protective skin barrier [42,43]. However, this parameter can be further analyzed in detail by using various criterion like evaluation of lipid content, formation of cornified envelope or by examining tight junctions. On the other hand, hyperproliferation can be revealed through the observation of proliferative markers such as Ki-67 and the proliferating cell nuclear antigen (PCNA). The increase of these nuclear proteins in psoriasis has already been shown in literature [23,44].

These local physical phenomena are accompanied by local inflammation that occurs through the dermal hypervascularity and angiogenesis which are characterized by the infiltration of blood cells in the skin [45]. Indeed, from day 4 an increase in the signal of CD31 and CD34, respectively a vascular and an endothelial marker, is observed in skin samples of IMQ-induced psoriatic mice. It has already been proven that epidermal keratinocytes and vascular endothelial cells are active participants in the psoriasis inflammatory process via secreted cytokines and growth factors along with the upregulation of signaling and adhesion molecules on their surfaces [16]. The changes in dermal vascularity have been reported in studies, however, the pathologic process behind these changes remain unclear and considerable evidence suggests the reason to be angiogenesis [37,46]. The increased expression of endothelial cells and vessels in our study are analogous to the above-mentioned studies and their expression is comparable to that of study conducted by Yazici et al. [44]. Furthermore, skin perfusion was also assessed using speckle and the redness in psoriatic skin was quantified (see Appendix A). Thus, these markers can be further used to determine the effect of anti-psoriatic drugs using IMQ-induced mouse model.

Systemic toxicity is also observed concomitantly with the onset of local signs of psoriasis. It has been illustrated that various types of cytokines are involved in the modulation of inflammatory responses in inflammatory skin diseases like psoriasis. Indeed, the cutaneous and systemic overexpression of various pro-inflammatory cytokines, such as interleukins and TNF-α, has been highlighted [47]. IL-17A is known to have a crucial role in the psoriasis’ pathogenesis: its skin expression is closely related to the development of skin lesions, with an increase starting after one day of IMQ application [4]. In this study there is not significant variation in IL-17A serum and skin levels which is not in accordance with the literature. These results can be explained by the low number of mice in each group (*n* = 3 for each day, *n* = 4 for control group) where inter-individual differences take precedence over the parameter fluctuation. A tendency to increase the level in the serum following days of IMQ application is still observed. However, the significant increase of TNF-α serum level at day 2 is correlated with the appearance of psoriasis phenotype and the important increase in spleen length. The systemic inflammation can be correlated with splenic inflammation as already described in patients with psoriasis [48]. Also, in the IMQ-induced mouse model the splenomegaly is reversed by the use of anti-psoriatic treatment, meaning that this phenomenon is not an IMQ side effect but it is due to the psoriasis itself [49]. We also hypothesized that the changes in the lymph nodes can be related to the severity of psoriasis. The results shown in our study can be supported by one that highlights the role of Tfh cell subsets in psoriasis [50]. Furthermore, it can be supported by the work that correlates the Tfh cells directly with the magnitude germinal centers and signify their role in immune mediated disorders [51,52]. Moreover, it has been proven by Yanaba et al. [53] that immune B cells via IL-10 production will then migrate to draining lymph nodes for their activation during IMQ-induced skin inflammation. Therefore, the appearance of germinal centers and the increase in area measurements of lymph nodes proves the usefulness of this biomarker for the future studies of IMQ-induced psoriasis.

To our knowledge, this is the first study that correlates the severity of psoriasis with the IMQ concentration in skin. After six days of IMQ application on mice back skin, total PASI score as well as the TEWL measurements reached a plateau indicating maximum skin barrier disruption. This conclusion is supported by the thickening of the epidermis and the appearance of pSTAT3 signal in the basal keratinocytes layer at day 6. Similarly, at the systemic level, the swelling of spleen and enlargement of lymph nodes are maximal at day 6, before the appearance of germinal centers at day 8. Thus, the IMQ concentration measured in skin, with highest values at day 6 and 8 is in correlation to the severity of the other symptoms (Table 1). This model sums up the first phase of the human biphasic psoriasis disease characterized by plaque type psoriasis lesions dominated by neutrophils infiltrates before the late phase with chronic plaques associated with Th1 response [54]. This mouse model needs a continuous application of IMQ in order to maintain the chronic inflammation which makes it a hard maintenance model because once the application of IMQ is stopped, there is a rapid return to normal phenotype (see Appendix A).

Skin is a semi-permeable barrier that discriminates molecules penetration based on their physico-chemical properties and size. As we described with this IMQ-induced psoriasis mouse model, psoriasis is a disease characterized by epidermal swelling and infiltration of inflammatory cells leading to the disruption of this barrier. These changes could improve the penetration of anti-psoriatic drugs either in the free form or through nano-encapsulation of topical treatment. This nano-vectorization method allows a lower drug concentration which can diminish undesirable effects and hence increase patient compliance [55]. Such strategies have already been developed and tested in IMQ-induced psoriasis mouse model [56]. In another internal project, tofacitinib was tested as JAK/STAT pathway inhibitor for the treatment of psoriasis, after incorporation into a cream and compared to Dermoval (GSK), a cream containing clobetasol propionate, commercialized against psoriasis lesions. Formulations were applied daily, twelve hours after the IMQ application on the presented mouse model. The non-appearance/resolution of the psoriatic phenotype on the basis of above mentioned psoriatic markers was taken in consideration to determine the treatment efficacy (see Appendix A).

The wide acceptability of this model cannot be ignored because of the various advantages that it offers. However, there is still a need to bridge the gaps that exist in understating human psoriasis using a comprehensive in vivo model. Hence, the parameters observed in this article can contribute to a comprehensive study of psoriatic characteristics using IMQ-induced mouse model and thus, can lead to a better preclinical evaluation of various pharmaceutical products.

## Figures and Tables

**Figure 1 pharmaceutics-12-00789-f001:**
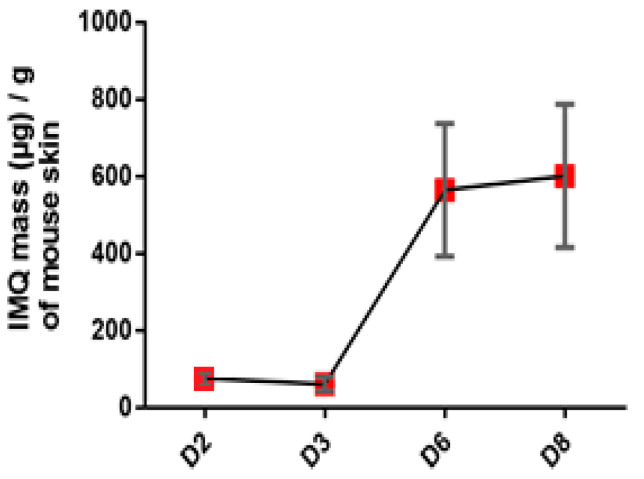
Evaluation of IMQ concentration in mouse skin in the course of IMQ-induced psoriasis-like dermatitis. Values are presented as mean ± SD.

**Figure 2 pharmaceutics-12-00789-f002:**
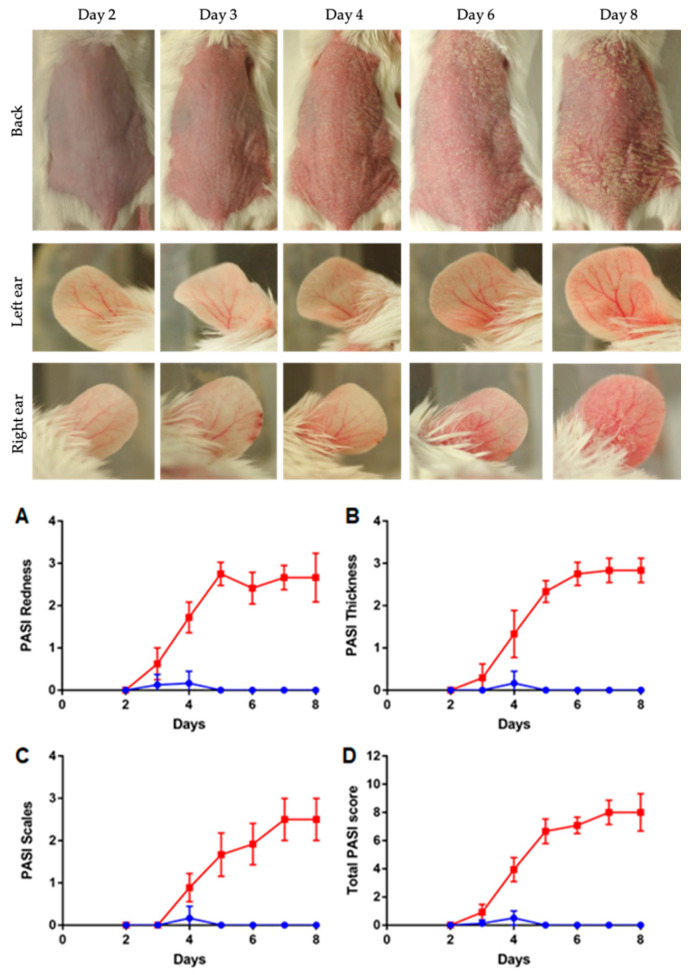
(**up**) Evaluation of dorsal skin and ears following IMQ cream application, from day 2 to day 8. Left ear was negative control, with no IMQ applied. (**down**) Evaluation of PASI score in the course of IMQ induction of psoriasis-like dermatitis. Observed criteria are skin redness (**A**), thickness (**B**), the presence of scales (**C**) and total PASI score (**D**). Psoriatic mice are presented with red squares, control mice with blue rounds. Data is presented as mean ± SD (*n* = 15 for day 1 and decreased subsequently to *n* = 3 for day 8).

**Figure 3 pharmaceutics-12-00789-f003:**
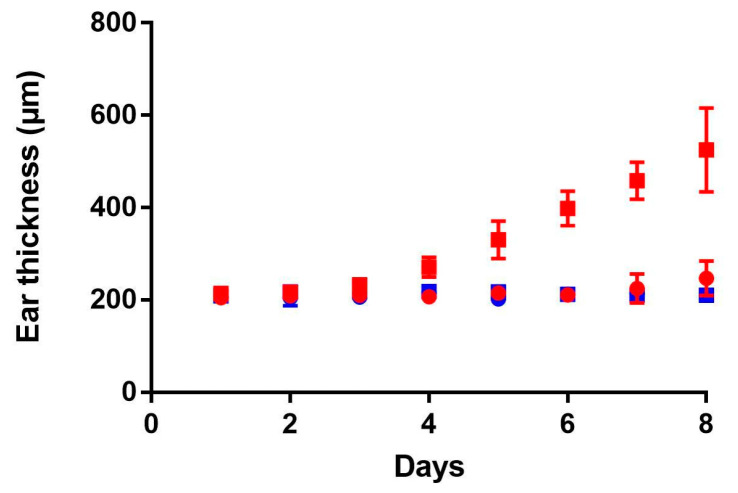
Evaluation of ear thickness during 8 days. Experimental group (daily IMQ application on the right ear) is shown in red squares and the control (left ear with Lanette application) is shown in red rounds; For control group, without any topical application, right ear is shown in blue squares and left ear in blue rounds. Data is presented as mean ± SD (*n* = 15 for day 1 and decreased subsequently to *n* = 3 for day 8).

**Figure 4 pharmaceutics-12-00789-f004:**
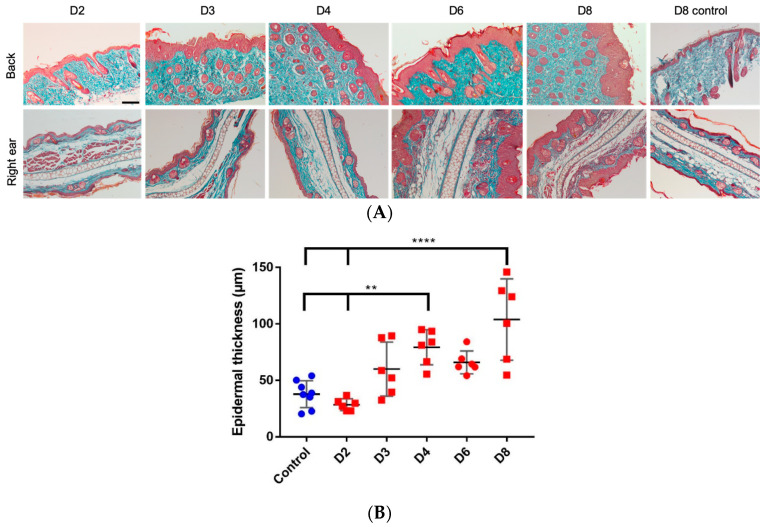
(**A**) Histological examination by Masson’s trichrome staining of back and right ear samples (scale bar = 100 µm for each image). (**B**) Measurement of epidermal back thickness for 8 days of IMQ application; measurements were made by two different persons on 3 images, for each day (*n* = 3) and statistically analyzed using one-way ANOVA followed by Tukey’s multiple comparison test. **: *p* < 0.01; ****: *p* < 0.0001.

**Figure 5 pharmaceutics-12-00789-f005:**
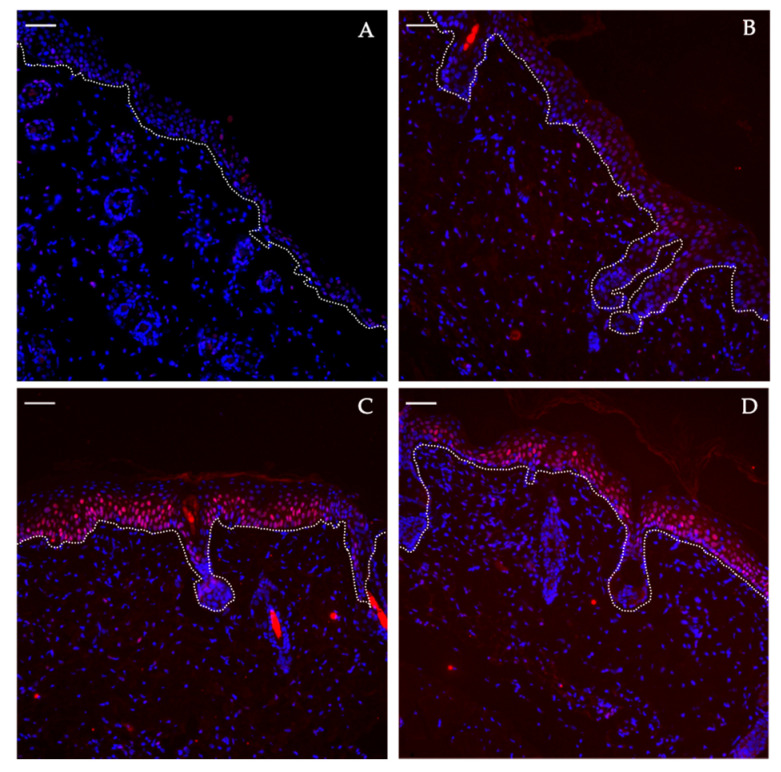
pSTAT3 staining evaluation on dorsal skin in the course of IMQ-induced psoriasis-like dermatitis at day 2 (**A**), day 4 (**B**), day 6 (**C**) and day 8 (**D**). The dotted white line represents the dermo-epidermal junction with epidermis upwardly. pSTAT3 protein appears in red and cell nuclei in blue. Scale bar = 50 µm.

**Figure 6 pharmaceutics-12-00789-f006:**
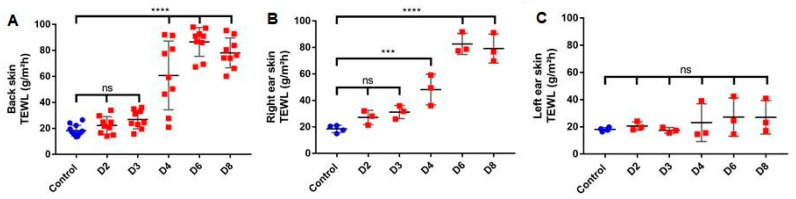
TEWL evaluation during IMQ induction of psoriasis-like dermatitis. TEWL was measured thrice at 3 spots for each mouse back (3 times on the left flank, 3 times on the right flank and 3 times on the upper back) (*n* = 3, except for control where *n* = 4) (**A**). TEWL was also measured on both ears, right ear with IMQ treatment (**B**) and left ear as control (**C**) (*n* = 3, except for control where *n* = 4). Results were statistically analyzed using one-way ANOVA followed by Dunnett’s multiple comparison test. Values are presented as mean ± SD. ***: *p* < 0.0005, ****: *p* < 0.0001.

**Figure 7 pharmaceutics-12-00789-f007:**
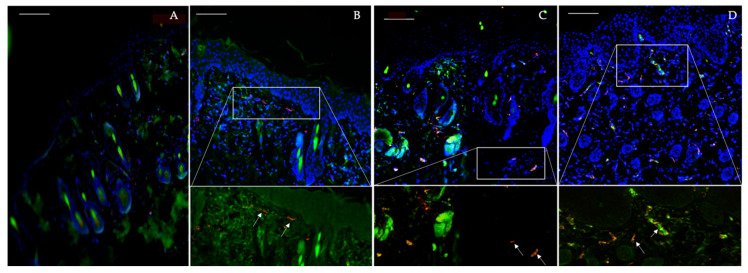
Endothelial markers evaluation in the course of IMQ-induced psoriasis-like dermatitis. Day 2 (**A**), Day 4 (**B**), Day 6 (**C**) and Day 8 (**D**). Blood vessels are presented in red indicating CD31, endothelial cells in green indicating CD34 and cell nuclei in blue by DAPI. Scale bar = 100 µm.

**Figure 8 pharmaceutics-12-00789-f008:**
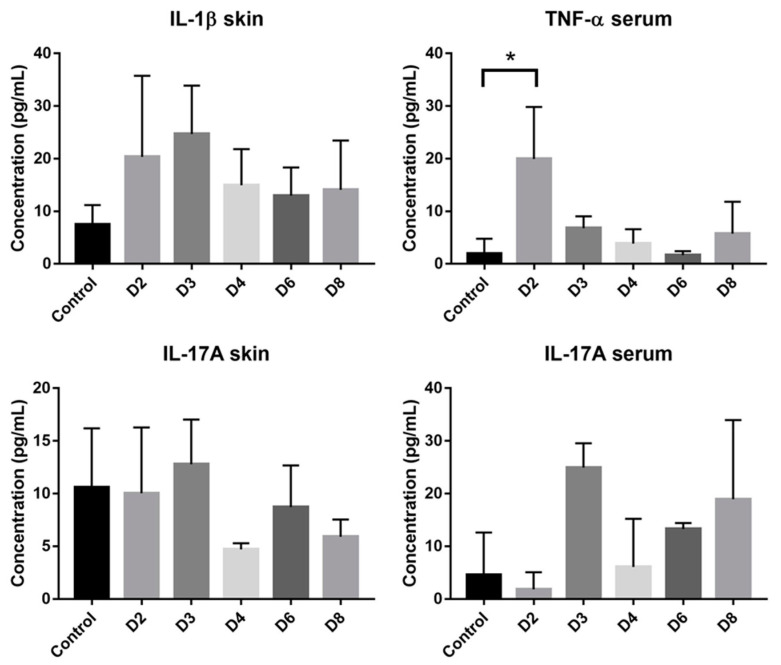
Cytokines concentration in both skin and serum samples. Data are presented as mean ± SD and statistically analyzed using one-way ANOVA followed by Tukey’s multiple comparison test. Only TNF-α level in serum at day 2 is significantly different from control. *: *p* < 0.05 (*n* = 3, except for control where *n* = 4).

**Figure 9 pharmaceutics-12-00789-f009:**
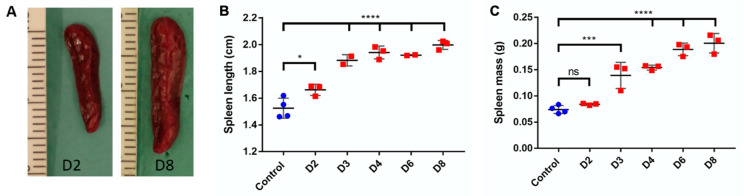
Evaluation of spleen length and mass in the course of IMQ-induced psoriasis-like dermatitis. Visual examination shows increased length and width of the spleen between days 2 and 8 (**A**). Spleen length (**B**) and mass (**C**) evaluations were analyzed using one-way ANOVA followed by Dunnett multiple comparison test. Values are presented as mean ± SD (*n* = 3, except for control where *n* = 4). *: *p* < 0.05, ***: *p* < 0.0005, ****: *p* < 0.0001.

**Figure 10 pharmaceutics-12-00789-f010:**
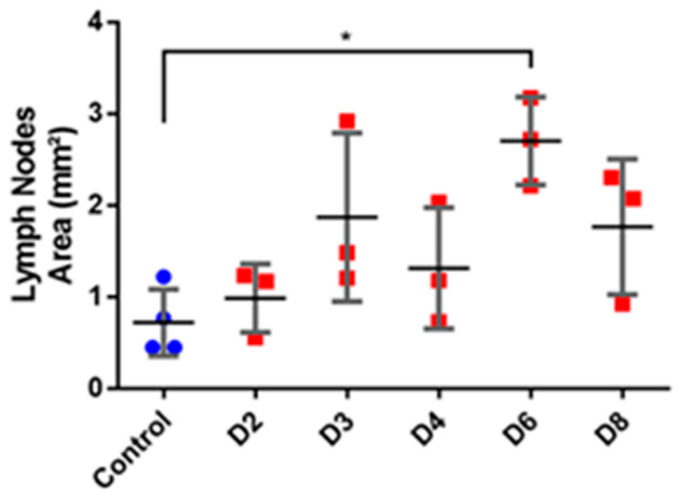
Evaluation of area of lymph nodes (mm^2^) in the course of IMQ-induced psoriasis-like dermatitis. Evaluations were analyzed using one-way ANOVA followed by Dunnett multiple comparison test. Values are presented as mean ± SD (*n* = 3, except for control where *n* = 4). *: *p* < 0.05.

**Figure 11 pharmaceutics-12-00789-f011:**
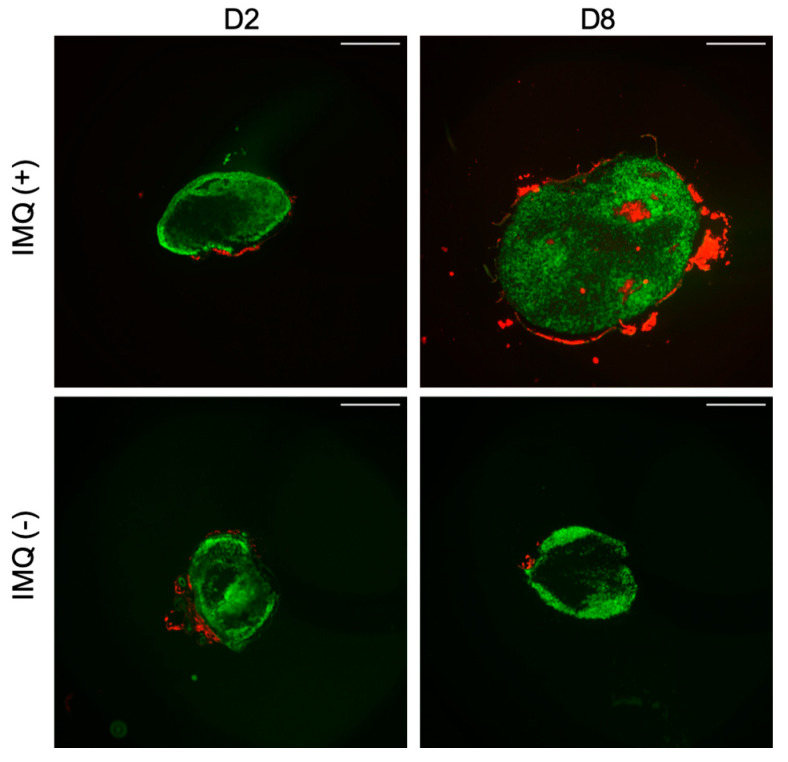
Immunofluorescent staining of germinal centers in mice inguinal lymph nodes. Sections were stained with anti-mouse IgD (BioLegend) to detect B cells follicles (green), and with biotinylated peanut agglutinin (PNA) (Vectors Laboratories) to detect germinal centers (red). Scale bar = 500 µm.

**Table 1 pharmaceutics-12-00789-t001:** Evolution of identified psoriatic parameters following the eight days of IMQ topical application (0: absence of parameter; -: basal level of the parameter; +/++/+++/++++: presence of the parameter with higher intensity than basal state, from + for weak increase until ++++ for high increase).

			Ctrl	D2	D3	D4	D6	D8
	IMQ concentration		0	+	+	+	++++	++++
Physical/ Histological Markers	PASI score		0	0	+	++	++++	++++
	Ear thickness		-	-	-	-	++	++++
	Epidermal thickness		-	-	+	++	++	++++
	pSTAT3 signal		0	0	0	0	++	++
	TEWL	Back	-	+	+	+++	++++	++++
		Ear	-	-	-	+	+++	+++
	Dermal vascularity		-	-	-	+	++	++
Systemic Toxicity	TNF-α serum		-	+	-	-	-	-
	Spleen	Length	-	+	+++	+++	+++	++++
		Mass	-	-	+	++	+++	+++
	Lymph nodes	Area	-	-	+	+	++	+
		Germinal Centers	0	0	0	0	0	+

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
