# Peer review of "Advanced Characterization of Imiquimod-Induced Psoriasis-Like Mouse Model"

_pharmaceutics, 2020, doi:10.3390/pharmaceutics12090789_

Round 1
Reviewer 1 Report
line 77: in my opinion, a greater explanation of the meaning of TEWL is needed. The author can be inspired by the various works in the literature, for example In vitro and in vivo trans-epidermal water loss evaluation following topical drug delivery systems application for pharmaceutical analysis. Cristiano, M.C., Froiio, F., Mancuso, A., Iannone, M., Fresta, M., Fiorito, S., Celia, C., Paolino, D. Journal of Pharmaceutical and Biomedical Analysis. Volume 186, 2020, Article number 113295.
Lines 96-99: This sentence needs bibliographic support.
Line 120: why did the authors choose to use that drug concentration? Please, explain it.
2.1 Animals: The journal requires specific information about the use of animals during experimentation. In particular Pharmaceutics’ guide for the author affirms “Manuscripts containing original descriptions of research conducted in experimental animals must contain details of approval by a properly constituted research ethics committee. As a minimum, the project identification code, date of approval and name of the ethics committee or institutional review board should be cited in the Methods section.” This important information seems to be missing.
3.2. Ear Thickness Measurements: in my opinion, the simply visual analysis is not suitable to characterize the effect of IMQ. In particular, about figure 2, the authors affirmed that “The application of IMQ for 8 consecutive days caused a significant increase in the ear thickness from day 6 due to inflammation (Figure 2A). Contrarily on the left ear, the control one without treatment, no change in ear thickness was observed for 8 days”; observing the figure, this different between untreated and treated ear is not so obvious. In particular, how do the authors explain the reduction of the left ear in D4, and its subsequent increase? Moreover, if D6 of right ear is analyzed in terms of negative standard deviation, the obtained value is below all others. These results highlight the inter-individual variables among the operators who conducted the analyzes.
3.7. Systemic Inflammatory Response: the authors described the evaluation of spleen length and mass consequently to the IMQ administration; are the authors sure that the increase in the spleen is due to induced psoriasis? can this increase simply be a side effect and secondary to the administration of the drug? in the latter case, the increase in spleen length and mass cannot be considered an indicator of psoriasis.
Have the authors tried to treat the skin of these animals characterized by psoriasis induced with anti-psoriasis treatments whose activity has already been confirmed? This test could be fundamental to evaluate if the alteration induced by IMQ could interfere with subsequent studies for which these animals would be useful.
Reviewer 2 Report
Jabeen et al present an interesting manuscript characterizing the effects of IMQ induced psoriasis on mice skin. This is one of the most commonly used models of psoriasis with significant amount of publications, therefore the novelty of these characterisations is low, although characterisation of lymph nodes in this model has not been previously reported to my knowledge.
There are number of suggestions that could significantly improve the manuscript as follows:
- Title is misleading, authors did not compare characterisation of the model in response to nanoparticle therapies, they simply provided advanced characterisation of the model. This needs to be clear both in title and conclusion.
- Use of ear thickness for epidermal swelling, TEWL and PASI score is not novel and has been done extensively in this model.
- Fig 1 suggest only presenting 1D (1A-C are not needed as its represented in total score in 1D. To increase novelty redness could be analysed using a hand held spectrometer.
- Fig 2A and 2B please use same scale on y-axis
- Fig 3A could be improved by measuring rate peg length using H&E sections. It is not clear why Masson Trichrome was used as this is generally a collagen stain, doesn't add any new info in this context/model as effects on collagen are not expected or analysed.
- Fig 4 authors report "intense" signal, this is very subjective but also images intensity need to be improved to justify pStat3 as valuable marker. Authors do not explain why they chose to look at pStat3 until discussion. And its not clear is this is the best marker for psoriasis pathways more detail is needed.
- Using this model number of authors have reported differences in skin and serum IL-17A levels however here authors do not observe this, can they explain why? Also it would be useful if some anti-inflammatory cytokines were also examined as comparison.
- It is not conventional to present 3 TEWL back readings in separate graphs, if cream is applied to entire back then no differences should be expected. Generally authors do 3 measurements on 3 areas of back but the average and present results in one graph. Fig6A-C should be combined in one graph.
- To increase novelty and characterisation of this model authors mention effects on skin barrier, this could be further supported by examining tight junctions (EM or WB of TJ proteins) or analysis of lipid content or formation of cornified envelope (percentage of round keratinocytes in a cornified envelope). Also this figure should be better as Fig 3 to improve flow of data.
- All figure legends should state the n.
- Figure 11 should be Fig 1 to improve flow of data.
- In discussion authors point to hyperproliferation as a key feature but did not asses it, eg using PCNA or KI67 staining.
- In discussion authors state IMQ "remains stable as long as psoriatic phenotype is in place" this needs rewording, it is the IMQ that causes the psoriatic phenotype, plus others papers have reported that phenotype quickly resolved once treatment is stopped, it would have been far more interesting if they characterised the resolution of phenotype once treatment is stopped as there is currently a high interest in understanding immunology of resolved psoriatic lesions that typically lead to disease relapse, and recurrent lesions in same location, and it is not clear if this model is useful for such studies, authors should at least comment on this in discission.
- Spleen data should have been supported with IL-10 analysis
- Last sentence of the discussion is not supported by data presented here, the authors did not compare drug efficacy analysis.
- Same for conclusion 3rd sentence please reword.
Reviewer 3 Report
The article fits the aims of the journal and falls within the journal purpose. The authors aim to better characterise a mice model of psoriasis-like dermatitis induced by topical 5% imiquimod (IMQ), a model that is already widely employed in research due to its increased similarities to human psoriatic plaque, judged from the perspective of epidermal, dermal and vascular components alterations. The assessment follows various indicators such as local aspect –erythema, induration, scaling; epidermal thickness; vascular endothelial markers; systemic inflammation; skin barrier alteration/disruption of skin barrier measured by transepidermal water loss (TEWL) and IMQ concentation measured by high performance liquid chromatography (HPLC).
The manuscript offers a deeper and somehow fresher perspective on the parameters governing this model, to a certain extent; however, it worth mentioning that the idea is not entirely novel, as these parameters have also been measured in imiquimod psoriasis model in mice in other studies beforehand, in various studies along time, each taking into consideration various parameters of the ones analysed here.
The article is an useful contribution to the journal and the research is well conducted; however, minor changesshould be taken into consideration:
Abstract: line 22: “dermal disease..” As psoriasis is a complex condition resulted from a very complex network of interactions between epidermal and dermal components, genetics and general factors, please rephrase “dermal disease”, as to comprehend the entire complexity of psoriasis, not confined to dermis, as it could be understood from the current phrasing.
Methodology. The methodologicaldesign is quite adequate.
Lines 80-84: while use of anti-CD31 is explained in the paragraph, a short insert to explicate the use of anti-CD34 staining would be useful to the reader.
Lines 245 to 247: statistical analysis- please insert manufacturer/countries of software used. Moreover, ANOVA is an omnibus statistical test, assesses whether a significant difference exists at all amongst the groups, overall, so, in order to make pairwise comparisons to check for differences properly chosen post-hoc tests may be used; although indeed stated in a figure legend (i.e. Tukey), please detail also in Methodology section what post hoc tests have been used/how pair-wise comparisons were made and how the authors mitigated the risk of overall multiple comparisons.
Nano-particles seem not to be the main focusof the article; it rather seems at most a pretext for the research or more of a not-that-close collateral possibility, as partly stated by authors, therefore it might be removed from the titleas it is rather a remote promise, confusing the reader at least to a certain extent, or, at least, the title might benefit from a certain degree of rephrasal, as to reflect clearly the scientific endevour.
A main focus for a revision should be correlating IMQ level and the others parameters, interleukins, epidermal thickness, pStat3, as well as other measured parameters. While it can be appreciated by eyeballing the graphs, atableto gather all masured parameters and relative correlations would be of great benefit to the reader, be it organised either in function of absorbed IMQ concentrations or in respect to days counted.
The pathology addressed is an important public health issue, as psoriasis is a frequent conditions in humans, with serious impact on patients’ quality of life. In this respect, a fuller characterisation of a mice mode is important, especially to further studies on various pharmaceuticals administration.
Grammar and punctuation must be carefully checked within the entire article.
The article is thoroughly documented and the scientific reasoning is sound; the study is well conducted.
Overall, I consider the article an useful contribution to the journal. Therefore, I recommend the manuscript for being published after suggested minor changes have been taken into consideration by the authors.
Round 2
Reviewer 1 Report
The answers are satisfactory enough
Author Response
The manuscript has been checked for English language by a native English speaker and necessary changes have been made. The corrections are highlighted in yellow in the latest version of manuscript.
Reviewer 2 Report
Authors have improved the flow and presentation of the results as well as readability. Both title and conclusion better reflect the work that was conducted now. The authors however did not add any extra experiments to support their data but have made textual changes.
Following should be addressed before this can be accepted:
- Lines 524 and 556 - data should be shown as it supports the characterisation of this model which is what the paper is trying to do to improve model characterisation for further research purposes
- Therapeutic efficacy of treatments mentioned on lines 564-577 should not be included or those claims removed unless the data is included to support these findings.
- Authors response to point 8 from letter response to reviewer comments needs to be discussed and highlighted in discussion as this is still a weak point.
Author Response
Point 1: Authors have improved the flow and presentation of the results as well as readability. Both title and conclusion better reflect the work that was conducted now. The authors however did not add any extra experiments to support their data but have made textual changes.
Response 1: To support the textual changes done in previous manuscript, the corresponding supplementary data has been provided. It includes: a) Speckle to study the skin perfusion of healthy and inflamed back skin; b) Experimental data indicating loss of phenotype after discontinuation of IMQ and c) Usefulness of IMQ induced mouse model to evaluate the efficacy of Tofacitinib.
Point 2: Lines 524 and 556 - data should be shown as it supports the characterisation of this model which is what the paper is trying to do to improve model characterisation for further research purposes
Response 2: In line 526, we discussed about skin perfusion measured through speckle, the materials and methods along with the results have been added in supplementary data (Figure S1).
In line 565, we discussed about the hard maintenance of this model as the phenotype disappears after the discontinuation of IMQ, we have shown these results in supplementary data (Figure S2).
Point 3: Therapeutic efficacy of treatments mentioned on lines 564-577 should not be included or those claims removed unless the data is included to support these findings.
Response 3: We have added the results proving the efficacy of JAK/STAT inhibitor Tofacitinib (free from) using IMQ induced model in supplementary data (Figure S3, A-D). We have removed the discussion about Ruxolitinib and JAK/STAT inhibitors in encapsulated form to avoid authorship conflict as this work was done in another internal project, in collaboration with another team member (line 574).
Point 4: Authors response to point 8 from letter response to reviewer comments needs to be discussed and highlighted in discussion as this is still a weak point.
Response 4: The lack of significant results in IL-17A variation in skin and serum was explained into the discussion. It is explained by the low number of mice in each group, but a tendency in accordance with Van der Fits el al., 2009 was observed (lines 532-539).
